# A Longitudinal Study of Chronic Periodontitis in Two Cohorts of Community-Dwelling Elderly Australians

**DOI:** 10.3390/ijerph191811824

**Published:** 2022-09-19

**Authors:** Xiangqun Ju, Jane Harford, Liana Luzzi, Gloria Mejia, Lisa M. Jamieson

**Affiliations:** 1Australian Research Centre for Population Oral Health (ARCPOH), Adelaide Dental School, University of Adelaide, Adelaide 5000, Australia; 2College of Nursing and Health Sciences, Flinders University, Adelaide 5042, Australia

**Keywords:** periodontitis, tooth loss, gingival recession (GR), probing pocket depth (PPD), clinical attachment level (CAL), incidence

## Abstract

**Background:** The study aimed to estimate and compare the incidence and progression of chronic periodontitis among two generations of older Australian adults. **Methods:** Data were from two population-based cohort studies of Australian older adults aged 60+ years South Australian Dental Longitudinal Studies (SADLS), SADLS I (1991–1992) and SADLS II (2013–2014). American Academy of Periodontology/the U.S. Centres for Disease Control and Prevention (CDC/AAP), and the 2018 European Federation of Periodontology classification (EFP/AAP) case definitions were used to define and calculate the incidence and progression of chronic periodontitis. Multivariable Poisson regression models were used to estimate incidence risk ratios (IRRs) of periodontitis. **Results:** A total 567 and 201 dentate respondents had periodontal exams in SADLS I and II, respectively. The incidence rate was greater in SADLS II than in SADLS I, approximately 200 vs. 100/1000 person years, respectively. Current smokers had more than two times higher IRRs, 2.38 (1.30–4.34) and 2.30 (1.24–4.26), than their non-smoking counterparts in the previous generation under the CDC/AAP and EFP/AAP, respectively. **Conclusions:** The most recent generation of older adults has greater incidence and progression of periodontitis than the previous generation. Being a current tobacco smoker was a significant risk factor for both the incidence and progression of periodontitis.

## 1. Introduction

Chronic periodontitis is highly prevalent in adults and older populations worldwide. According to the Global Burden of Disease 2015, more than 7% of the world’s population, approximately 540 million people, have severe chronic periodontitis [1]. Chronic periodontitis is characterised by gingival recession and/or periodontal pocket formation due to the destruction of periodontal tissues and sometimes alveolar bone [2,3].

Although tooth loss has decreased in recent decades [4], chronic periodontitis continues to be one of the major causes of tooth loss. It is estimated that 30% to 35% of all tooth extractions are attributed to periodontitis [1]. The number of teeth lost, as well as tooth loss patterns and type, impact masticatory function [5]. This, in turn, may lead to decreased intake of nutrients which impact the immune system. Periodontitis is associated with systemic diseases, such as diabetes mellitus [6], cardiovascular disease, respiratory disease, renal disease, obesity, osteoporosis and cancer [7,8].

Evidence suggests that contemporary older adults, by virtue of increased life expectancy and retention of teeth, are susceptible to chronic periodontitis [4,9]. In Australia, chronic periodontitis affected more than 60% and 70% of older Australians in national oral health surveys conducted in 2004–2006 and 2017–2018, respectively [10,11]. It is necessary to estimate the incidence and progression chronic periodontitis to understand, at a population level, how disease patterns change over time, and the risk factors. However, longitudinal data on the incidence and progression of periodontitis are scarce. The aim of this study was to compare the incidence and progression of chronic periodontitis among two generations of the elderly Australian adults using longitudinal data. We hypothesized that the incidence and progression of periodontitis would be higher among the most recent older generation than the previous generation.

## 2. Methods

This study is reported according to STROBE (Strengthening the Reporting of Observational Studies in Epidemiology) guidelines.

### 2.1. Study Design, Setting and Participants

This cohort study draws data from two population-based longitudinal surveys: (1) the South Australian Dental Longitudinal Study (SADLSI) with baseline information gathered in 1991–1992 [12] and (2) Intergenerational change in oral health in Australia (SADLS II) with baseline data in 2013–2014. Both studies had a two-year follow-up (in 1993–1994 and 2015–2016, respectively).

Study participants represent the generation of older Australians born before 1931 (SADLS I) and 1953 (SADLS II), which hereafter will be named as previous and recent generations, respectively. Stratified random samples of non-institutionalised people aged 60 years or over, residing in the capital city of Adelaide and the regional city of Mount Gambier were selected from the South Australian State Electoral Database in 1991 and 2013, respectively. All participants provided signed informed consent.

### 2.2. Data Collection

Data collection included an interview and oral clinical examination. A face-to-face questionnaire interview, including social demographic characteristics, general health, and oral health-related behaviours, was conducted by a trained interviewer. A standardised oral epidemiological examination was conducted by registered and calibrated dentists (including four and three dentists in SADLS I and II, respectively) for dentate participants. All teeth present in the mouth (including third molars) were assessed based on the U.S National Institute of Dental Research survey of employed adults and seniors [13].

The periodontal assessment included gingival recession (GR) and probing pocket depth (PPD) measurements at mesio-buccal, mid-buccal, and disto-lingual sites of each tooth present. GR measurements were recorded as positive when recession was present, and a negative value for GR was recorded where the gingival margin was located more than 1 mm coronally to the cementoenamel junction. Clinical attachment level (CAL) was computed during data analysis through the sum of GR and PPD. Inter-examiner reliability was assessed through replicate examinations of 28 and 29 Australian older adults in SADLS I [12] and SADLS II, respectively. Intra-class correlation coefficients for the periodontal measurements were 0.46 and 0.73 for mean PPD, 0.92 and 0.90 for mean GR and 0.84 and 0.75 for mean CAL in SADLS I [12] and SADLS II, respectively, indicating medium-to-excellent reliability.

### 2.3. Outcome Variables

The outcome variable was combined incidence or progression of chronic periodontitis (Yes vs. No) at two-year follow-up, to measure declining periodontal health (the worsening of disease).

Chronic periodontitis was assessed based on two periodontal case definitions: (1) The American Association of Periodontology and the U.S. Centres for Disease Control and Prevention case definition (AAP/CDC) [14], and (2) The 2018 European Federation of Periodontology/American Academy of Periodontology classification (EFP/AAP) [2] to describe the degree of chronic periodontitis in the two health surveys (Table 1).
The incidence of periodontitis (new cases) was defined from the baseline to two-year follow-up in the two studies as from no indication of periodontitis (‘No disease’) to some indication of periodontitis, such as: (1) from none to mild, moderate or severe periodontitis under the AAP/CDC case definition; and (2) from No/gingivitis to stage I, Stage II or Stage III–IV under the EFP/AAP case definition.The progression of periodontitis was defined from baseline to two-year follow-up in the two studies as: (1) from mild to moderate or severe, or from moderate to severe to tooth loss due to periodontitis under AAP/CDC case definition; and (2) from Stage I to Stage II, III or IV to tooth loss due to periodontitis under the EFP/AAP case definition.

### 2.4. Covariates

Covariates included baseline sociodemographic characteristics, oral health and related behaviours and general health.

Sociodemographic characteristics were ‘Age’, categorised into six groups (60–64, 65–69, 70–74, 75–79, 80–84, or ≥85 years); ‘Sex’ (Male vs. Female); ‘Marital status’ (‘Married/De-facto’ vs. ‘Single’); ‘Born in Australia’ (Yes vs. No); ‘Highest educational attainment’ (‘Secondary school’, ‘Trade to Diploma degree’ or ‘Tertiary’); and ‘Household income’ (‘Low (<AUD 20,000)’, ‘Medium (AUD 20,000 to 50,000)’ or ‘High (>AUD 50,000)’).

Health behaviours included ‘Tobacco smoking status’ (‘Current smoker’, ‘Used to smoke’ or ‘Never smoked’); ‘Alcohol drinking’ (‘Yes’ vs. ‘No’); and ‘Dental insurance statuses’ (‘Had’ vs. ‘No’). Oral health-related behaviours included frequency of teeth brushing, last dental visit, and usual reason for dental visit, and were dichotomised into ‘≥twice/day’ vs. ‘<twice/day’, ‘≤12 months’ vs. ‘>12 months’; and ‘Check-up’ vs. ‘Problem’, respectively.

General health was measured by self-reported histories of 10 chronic diseases, which included asthma, arthritis, cancer or malignancy, cataracts, chronic obstructive pulmonary disease (combining chronic bronchitis and/or emphysema (COPD)), diabetes, hypertension or high blood pressure, heart condition or heart attack, stroke or a small stroke (TIA), or/and osteoporosis or hip fracture, and was dichotomised into ‘Yes’ or ‘No’.

### 2.5. Statistical Analyses

Only the respondents who had the periodontal assessment at baseline and repeated at two-year follow-up were included in the analysis. Basic descriptive analyses were conducted to ascertain frequencies of the sample characteristics, and incidence rate of periodontitis. Multivariable Possion regression models with robust standard error estimation were generated. Unadjusted and adjusted incidence rate ratios (IRRs) and 95% CI were calculated for incidence and progression of periodontitis. Model 1 was adjusted for social demographic characteristics; Model 2 was model 1 plus adjustment for dental health related behaviours; and Model 3 was Model 2 plus adjustments for general health characteristics.

Data files were managed and summary variables computed using SAS software version 9.4 (SAS 9.4, SAS Institute Inc., Cary, NC, USA).

## 3. Results

A total of 1650 and 810 dentate participants were interviewed in SADLS I and SADLSII, respectively. Of those, 567 and 201 persons, respectively, had periodontal exams at baseline and two-year follow-up and were included the final data analysis. At baseline, the average age was 70.6 (SD = 7.26) years in SADLS I and 69.4 (SD = 6.5) years in SADLS II.

The majority of baseline characteristics were similar between those who were lost to follow-up and those who were not (see Appendix A), except for age and dental visiting status in SADLS I, and country of birth in SADLS II. A higher prevalence of those were lost to follow-up had diabetes and chronic diseases in SADLS I (see Appendix A), No statistically significant differences were noted between those followed up and lost to follow-up in SADLS II.

Table 2 presents the baseline sample characterisers of SADLS I and II participants. The proportion of age groups, gender, marital status and country of birth were similar cross the two cohorts, with more than half in the younger age group (60–69 years) and male, and approximately one quarter being ‘Single’ and ‘Born overseas’. The proportion of participants with the highest educational level (Tertiary) was higher, but the middle education group (Trade/diploma) was higher in SADLS I than in SADLS II, with the difference being around ten percent. The proportion with high household income and having dental insurance were lower in SADLS I than in SADLS II, from around one-fourth to more than one-third, and from more than forty to sixty percent, respectively. Both alcohol consumption and the proportion of participants reporting they had never smoked were higher (approximately 25 percent) in SADLS I than in SADLS II. Dental behaviours (oral hygiene and dental visiting) was similar between the two cohorts.

Table 3 shows ten chronic diseases among Australian older adults in the two surveys. The proportion with diabetes, hypertension and osteoporosis was higher in SADLS II than in SADLS I, while the proportion with heart disease decreased by almost 55 percent.

The mean number of teeth lost increased slightly at the two-year follow-up in both surveys, although there was a decrease of approximately 13 to 7 teeth across the two generations. The prevalence of severe periodontitis decreased with increasing tooth loss (see Appendix A), which implied the progression of periodontitis.

Figure 1 depicts the rate of chronic periodontitis (incidence + progression) among Australian older adults across the 22-year study period. A similar trend was observed under the AAP/CDC and EFP/AAP case definitions: the incidence rate was highest in SADLS II (more than 210/per 1000 person-year) than in SADLS I (approximately 100/per 1000 person-year).

Table 4 shows the bivariate analysis of the incidence and progression of periodontitis across two generations under the two case definitions. Higher IRRs were observed among males, current and previous smokers, those with cataracts in SADLS I, and experience of stroke in SADLS II.

After adjusting for sociodemographic characteristics, males had 1.5 times higher IRRs than female older Australian adults; after adjusting for all covariates, current smokers had 2 times higher IRRs than their non-smoking counterparts in SADLS I under the two case definitions (Appendix A). There were no significant differences observed in SADLS II, except the middle age group (70–74 years), which presented lower IRRs (0.3) than the youngest age group (60–64 years) (Appendix A).

## 4. Discussion

To the best of our knowledge, this is the first study to examine the incidence and progression of periodontitis among two older generations in Australia. Our findings indicated that the incidence and progression of periodontitis was higher in the most recent generation than in the previous generation, which supported our hypothesis. Incidence and progression of periodontitis was associated with tobacco smoking, especially in the older generation.

Higher incidence and progression of periodontitis higher among the most recent generation, which is likely attributed to the greater number of teeth retained and possibly the higher prevalence of periodontitis-associated chronic diseases [8], such as diabetes [6] and hypertension [15]. Given that periodontal disease is influenced by social determinants, the lower proportion of participants in the higher education level (Tertiary) [16] could also contribute to this greater prevalence of periodontal disease.

Periodontal therapy is an important step to prevent further disease progression, to reduce the risk of tooth loss, and to improve overall periodontal health. Evidence also suggests a bidirectional relationship between periodontitis and type 2 diabetes (T2D) [6]. Periodontal treatment can reduce glycated haemoglobin (HbA1c) levels, and possibly decrease the risk of death related to T2D [17]. Previous studies [18,19,20,21] have shown that in additional to controlling plaque and reducing probing depth and attachment loss, periodontal treatment can contribute to the overall control of oral and systemic inflammation (such as decreasing level of C-Reactive Protein and increasing level of IL-1, IL6 and TNF-α in blood). Given this study included a baseline measurement of diabetes, our findings can corroborate evidence that diabetes is a significant risk factor associated with the incidence/progression of chronic periodontitis. However, it is beyond the scope of this study to examine whether periodontal therapy contributed to a lessening of the effects of diabetes.

The association between tobacco smoking and incidence and progression of periodontitis in SADLS I is consistent with a systematic review [22] in which the authors conclude that tobacco smoking has a detrimental effect on the incidence and progression of periodontitis. This association was not observed in SADLS II, which might be attributed to the smaller proportion of participants who were current or former smokers.

This study has a number of strengths. To our knowledge, it is the first time that incidence and progression of periodontitis has been estimated across two generations of elderly Australians using longitudinal data. The study used two common periodontal case definitions, which increased the reliability of our findings. Finally, the inclusion of tooth loss due to periodontitis as part of the progression improved the accuracy of estimating incidence and progression of periodontitis.

Loss to follow-up is a limitation of this study. Loss to follow-up is inevitable in longitudinal studies and has the potential for bias. We compared baseline characteristics of participants remaining in the study and those lost to follow-up in the two cohorts and found no statistically significant differences for the majority of characteristics. In addition, the random sample collection and high response rates for the interview (around 65%) and oral examination (about 75%) at baseline, and good follow-up rates (more than 70% in SADLS I and nearly 60% in SADLS II) mitigated the potential for bias. The results are representative of only the study sample and cannot be generalised to the broader population. Another limitation is that periodontal status was measured at three sites, rather than six sites of each tooth, ref. [23] which might have clinical implications such as the potential for an underestimation of periodontitis. In future research, the staging and grade of periodontal disease should be based on the most recent recommendations from the United States [2], adopting a causal modelling approach to assess the effects of periodontitis risk factors (such as the change in systemic disease status, functional limitation and cognitive impairment) and of periodontal treatment on the incidence and progression of periodontitis. This could provide the necessary evidence to prevent and reduce the incidence of periodontitis and tooth loss and contribute to improving people’s quality of life.

## 5. Conclusions

Our findings indicated that the most recent generation (SADLS II) of older adults has higher incidence and progression of chronic periodontitis than the previous generation (SADLSI), despite the lower number of missing teeth. Independent of the generation studied, tobacco smoking is a significant risk factor for the incidence and progression of periodontitis.

## Figures and Tables

**Figure 1 ijerph-19-11824-f001:**
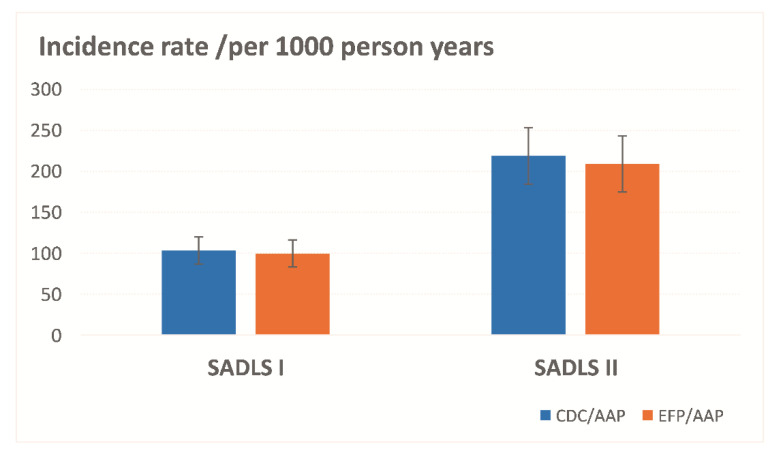
Incidence rate of chronic periodontitis among Australian adults in SADLS I and II under the AAP/CDC and EFP/CDC case definitions.

**Table 1 ijerph-19-11824-t001:** Different periodontal disease case definitions.

AAP/CDC		EFP/AAP	
The American Association of Periodontology and the U.S. Centres for Disease Control and Prevention Case Definition [14]		The 2018 European Federation of Periodontology/American Academy of Periodontology Classification [2]	
None/mild	Mild periodontitis is the presence of either two sites between adjacent teeth where CAL < 4 mm and PD < 5 mm.	Stage I	Periodontitis severity stage I is the presence of each tooth where 1 mm ≤ CAL ≤ 2 mm; no tooth loss due to periodontitis.
Moderate	Moderate periodontitis is the presence of either two sites between adjacent teeth where 4 mm ≤ CAL ≤ 6mm or at least two such sites have PD ≥ 5 mm.	Stage II	Periodontitis severity stage II is the presence of each tooth where 3 mm ≤ CAL ≤ 4mm; no tooth loss due to periodontitis.
Severe	Severe periodontitis is at least two sites between adjacent teeth where CAL ≥ 6 mm and there is at least one site PPD ≥ 5 mm.	Stage III–IV	Periodontitis severity stage III-IV is the presence of each tooth where CAL ≥ 5 mm or there is at last one site PPD ≥ 6 mm; or had tooth loss due to periodontitis of ≥4 teeth.

**Table 2 ijerph-19-11824-t002:** Sample characterisers of two cohort of older Australian adults at baseline.

	SADLS I (1991–1992) (*n* = 567)		SADLS II (2013–2014) (*n* = 201)	
	N	95% CI	N	95% CI
Sample demographic characteristics				
Age groups				
60–64	157	**27.7 (24.0–31.4)**	54	**27.4 (21.1–33.7)**
65–69	153	**27.0 (23.3–30.6)**	65	33.0 (26.4–39.6)
70–74	94	**16.6 (13.5–19.6)**	41	**20.8 (15.1–26.5)**
75–79	101	**17.8 (14.7–21.0)**	21	**10.7 (6.3–15.0)**
80–84	49	**8.6 (6.3–11.0)**	12	**6.1 (2.7–9.5)**
≥85	13	**2.3 (1.1–3.5)**	4	**2.0 (0.0–4.0)**
Sex				
Female	227	**40.0 (36.0–44.1)**	91	46.0 (39.0–53.0)
Male	340	**60.0 (55.9–64.0)**	107	54.0 (47.0–61.0)
Married status				
Married/De-facto	420	74.1 (70.5–77.7)	148	75.1 (69.0–81.2)
Single	147	25.9 (22.3–29.5)	49	24.8 (18.8–31.0)
Born in Australia				
Yes	387	**68.4 (64.5–72.2)**	143	**74.5 (68.3–80.7)**
No	179	**31.6 (27.8–35.5)**	49	**25.5 (19.3–31.7)**
Education level				
Tertiary	193	**34.1 (30.2–38.1)**	36	**18.3 (12.8–23.7)**
Trade/diploma degree	113	**20.0 (16.7–23.3)**	61	**31.0 (24.5–37.5)**
Secondary	260	**45.9 (41.8–50.1)**	100	**50.8 (43.7–57.8)**
Household income				
High	137	25.6 (21.9–29.3)	64	37.4 (30.1–44.8)
Medium	177	**33.0 (29.0–37.0)**	49	28.7 (21.8–35.5)
Low	222	**41.4 (37.2–45.6)**	58	33.9 (26.8–41.1)
Oral health-related behaviours				
Dental insured				
Had	244	**43.4 (39.3–47.5)**	126	**63.3 (56.6–70.1)**
No	318	**56.6 (52.5–60.7)**	73	**36.7 (29.9–43.4)**
Smoke status				
Never smoked	252	44.9 (40.8–49.0)	116	**59.2 (52.2–66.1)**
Used smoker	245	**43.7 (39.6–47.8)**	72	**36.7 (29.9–43.5)**
Current smoker	64	**11.4 (8.8–14.0)**	8	**4.1 (1.3–6.9)**
Alcohol drinking				
No	136	**24.0 (39.3–47.5)**	36	**18.3 (12.8–23.7)**
Yes	430	**76.0 (72.4–79.5)**	161	**81.7 (76.3–87.2)**
Dental behaviours				
Oral hygiene (Tooth brushing)				
At least twice/day	361	**63.7 (59.7–67.6)**	138	**69.3 (62.9–75.8)**
Less than twice/day	206	**36.3 (32.4–40.3)**	61	**30.7 (24.2–37.1)**
Last dental visiting				
Less than 12 months	360	**63.7 (59.7–67.7)**	139	**70.2 (63.8–76.6)**
More than 12 months	205	**36.3 (32.3–40.3)**	59	**29.8 (23.4–36.2)**
Reasons for dental visiting				
Check	252	**44.7 (40.7–48.8)**	89	46.1 (39.0–53.2)
Problem	312	**55.3 (51.2–59.4)**	104	53.9 (46.8–61.0)

**Note:** Difference statistically significant as denoted by non-over-lapping 95% confidence intervals (Bold).

**Table 3 ijerph-19-11824-t003:** Chronic diseases of two cohort of Australian older adults at baseline.

General Health	SADLS I (1991–1992) (*n* = 567)		SADLS II (2013–2014) (*n* = 201)	
	N	95% CI	N	95% CI
Asthma				
No	525	**92.6 (90.4–94.8)**	165	**86.8 (82.0–91.7)**
Yes	42	**7.4 (5.2–9.6)**	25	**13.2 (8.3–18.0)**
Arthritis				
No	280	49.5 (45.3–53.6)	97	51.9 (44.6–59.1)
Yes	286	50.5 (46.4–54.7)	90	48.1 (40.9–55.4)
Cancer				
No	502	**88.8 (86.2–91.5)**	159	**84.6 (79.4–89.8)**
Yes	63	**11.2 (8.5–13.8)**	29	**15.4 (10.2–20.6)**
Cataracts				
No	472	**83.2 (80.2–86.3)**	147	**80.3 (74.5–86.1)**
Yes	95	**16.8 (13.7–19.8)**	36	**19.7 (13.9–25.5)**
COPD				
No	505	**89.2 (86.7–91.8)**	167	**90.8 (86.5–95.0)**
Yes	61	**10.8 (8.2–13.3)**	17	**9.2 (5.0–13.5)**
Diabetes				
No	542	**95.6 (93.9–97.3)**	161	**85.2 (80.1–90.3)**
Yes	25	**4.4 (2.7–6.1)**	28	**14.8 (9.7–19.9)**
Hypertension				
No	365	**64.4 (60.4–68.3)**	94	49.5 (42.3–56.6)
Yes	202	**35.6 (31.7–39.6)**	96	50.5 (43.4–57.7)
Heart diseases				
No	465	**82.3 (79.1–85.5)**	170	**91.9 (87.9–95.9)**
Yes	100	**17.7 (14.5–20.9)**	15	**8.1 (4.1–12.1)**
Osteoporosis or hip fracture				
No	546	**96.5 (94.4–98.0)**	168	89.8 (85.5–94.2)
Yes	20	**3.5 (2.0–5.1)**	19	10.2 (5.8–14.5)
Stroke				
No	529	93.5 (91.4–95.5)	182	**97.3 (95.0–99.7)**
Yes	37	6.5 (4.5–8.6)	5	**2.7 (0.3–5.0)**
Chronic diseases (At least one)				
No	123	**21.7 (18.3–25.1)**	29	**14.7 (9.7–19.7)**
Had	444	**78.3 (74.9–81.7)**	168	**85.3 (80.3–90.3)**
Number of diseases				
0	123	21.7 (18.3–25.1)	29	14.7 (9.7–19.7)
1	163	28.7 (25.0–32.5)	62	31.5 (24.9–38.0)
2	151	26.6 (23.0–30.3)	53	26.9 (20.7–33.2)
3	79	13.9 (11.1–16.8)	30	15.2 (10.2–20.3)
4	34	6.0 (4.0–8.0)	15	7.6 (3.9–11.4)
5	10	1.8 (0.7–2.9)	6	3.0 (0.6–5.5)
6	6	1.1 (0.2–1.9)	2	1.0 (0.0–2.4)
7	1	0.2 (0.0–0.5)	0	0.0 (0.0–0.0)

**Note:** Difference statistically significant as denoted by non-over-lapping 95% confidence intervals (Bold).

**Table 4 ijerph-19-11824-t004:** Bivariate analysis of the incidence rate ratios (IRRs) of periodontitis under two case definition.

	SADLS I (*n* = 567)		SADLS II (*n* = 201)	
	CDC/AAP	EFP/AAP	CDC/AAP	EFP/AAP
	IRR (95% CI)	IRR (95% CI)	IRR (95% CI)	IRR (95% CI)
**Sample demographic characteristics**				
**Age groups**				
60–64	ref	ref	ref	ref
65–69	0.87 (0.57–1.35)	1.19 (0.76–1.87)	1.01 (0.69–1.49)	1.22 (0.79–1.90)
70–74	0.98 (0.61–1.59)	1.22 (0.74–2.02)	0.68 (0.40–1.17)	0.92 (0.53–1.60)
75–79	1.01 (0.63–1.60)	1.31 (0.80–2.12)	0.98 (0.57–1.69)	1.35 (0.78–2.40)
80–84	0.87 (0.46–1.63)	1.09 (0.56–2.07)	0.90 (0.43–1.86)	1.13 (0.53–2.39)
≥85	1.37 (0.57–3.28)	1.27 (0.44–3.66)	0.54 (0.10–3.02)	0.68 (0.12–3.82)
**Sex**				
Female	ref	ref	ref	ref
Male	**1.70 (1.19–2.40)**	**1.44 (1.02–2.05)**	0.87 (0.63–1.20)	0.87 (0.62–1.21)
**Married status**				
Married/De-facto	ref	ref	ref	ref
Single	0.79 (0.54–1.17)	0.95 (0.65–1.37)	0.80 (0.53–1.21)	0.91 (0.61–1.37)
**Born in Australia**				
Yes	ref	ref	ref	ref
No	1.33 (0.97–1.84)	1.34 (0.98–1.88)	1.01 (0.70–1.45)	1.07 (0.74–1.55)
**Education level**				
Tertiary	ref	ref	ref	ref
Trade/diploma degree	0.92 (0.58–1.47)	0.73 (0.43–1.23)	0.96 (0.62–1.51)	1.00 (0.64–1.56)
Secondary	1.06 (0.75–1.51)	1.12 (0.79–1.60)	0.88 (0.58–1.35)	0.78 (0.50–1.20)
**Household income**				
High	ref	ref	ref	ref
Medium	1.14 (0.79–1.64)	1.11 (0.76–1.62)	0.93 (0.60–1.44)	1.04 (0.66–1.65)
Low	1.00 (0.66–1.51)	1.08 (0.72–1.62)	0.87 (0.57–1.34)	0.94 (0.60–1.47)
**Oral health and related behaviours**				
**Dental insured**				
Had	ref	ref	ref	ref
No	1.07 (0.78–1.48)	1.17 (0.84–1.62)	1.05 (0.75–1.45)	0.92 (0.65–1.31)
**Smoke status**				
Never smoked	ref	ref	ref	ref
Used smoker	**1.71 (1.19–2.46)**	**1.54 (1.06–2.22)**	0.83 (0.58–1.18)	0.74 (0.50–1.08)
Current smoker	**2.18 (1.39–3.44)**	**1.94 (1.21–3.10)**	1.23 (0.66–2.27)	0.98 (0.46–2.09)
**Alcohol drinking**				
No	ref	ref	ref	ref
Yes	1.44 (0.95–2.18)	1.19 (0.80–1.76)	1.05 (0.68–1.60)	0.99 (0.64–1.51)
**Oral hygiene (Tooth brushing)**				
At least twice/day	ref	ref	ref	ref
Less than twice/day	1.19 (0.86–1.63)	1.10 (0.79–1.52)	1.22 (0.89–1.68)	1.25 (0.89–1.73)
**Last dental visiting**				
Less than 12 months	ref	ref	ref	ref
More than 12 months	0.84 (0.60–1.18)	0.84 (0.60–1.18)	1.13 (0.81–1.57)	0.91 (0.63–1.32)
**Reasons for dental visiting**				
Check	ref	ref	ref	ref
Problem	1.01 (0.74–1.39)	1.09 (0.79–1.51)	0.93 (0.67–1.29)	0.93 (0.66–1.31)
**General health**				
Asthma				
No	ref	ref	ref	ref
Yes	1.29 (0.76–2.22)	1.21 (0.69–2.14)	0.94 (0.57–1.57)	1.12 (0.69–1.81)
**Arthritis**				
No	ref	ref	ref	ref
Yes	0.96 (0.70–1.33)	1.11 (0.8–1.54)	0.98 (0.71–1.35)	0.88 (0.62–1.23)
**Cancer**				
No	ref	ref	ref	ref
Yes	1.26 (0.80–2.00)	1.33 (0.83–2.08)	1.22 (0.83–1.82)	1.20 (0.79–1.83)
**Cataracts**				
No	ref	ref	ref	ref
Yes	**1.49 (1.03–2.16)**	1.00 (0.64–1.56)	1.02 (0.68–1.54)	1.02 (0.66–1.57)
**COPD**				
No	ref	ref	ref	ref
Yes	0.77 (0.43–1.40)	0.98 (0.58–1.68)	1.09 (0.64–1.86)	1.16 (0.68–1.98)
**Diabetes**				
No	ref	ref	ref	ref
Yes	1.59 (0.88–2.89)	1.00 (0.46–2.24)	1.61 (0.87–2.96)	**2.09 (1.01–4.33)**
**Hypertension**				
No	ref	ref	ref	ref
Yes	0.94 (0.67–1.32)	0.92 (0.64–1.30)	0.83 (0.60–1.14)	0.91 (0.65–1.27)
**Heart disease**				
No	ref	ref	ref	ref
Yes	1.39 (0.96–2.02)	1.34 (0.91–1.97)	0.60 (0.25–1.40)	0.80 (0.38–1.67)
**Osteoporosis or hip fracture**				
No	ref	ref	ref	ref
Yes	0.24 (0.03–1.60)	0.24 (0.04–1.66)	1.25 (0.78–1.98)	1.01 (0.58–1.76)
**Stroke**				
No	ref	ref	ref	ref
Yes	1.50 (0.89–2.53)	0.95 (0.48–1.90)	**1.87 (1.17–2.98)**	**1.97 (1.23–3.15)**
**Chronic diseases (At least one)**				
No	ref	ref	ref	ref
Had	1.07 (0.72–1.64)	0.77 (0.53–1.11)	1.06 (0.67–1.70)	1.24 (0.73–2.11)

Note: Bold used for the statistically significant.

## Data Availability

The datasets generated and/or analysed during the current study are not publicly available due to privacy issues of the participants. Data are available from the corresponding author on reasonable request.

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
