# Peer review of "A Longitudinal Study of Chronic Periodontitis in Two Cohorts of Community-Dwelling Elderly Australians"

_ijerph, 2022, doi:10.3390/ijerph191811824_

Round 1

Reviewer 1 Report

This is an interesting read, although concerning that incidence and progression of chronic periodontitis is increasing with more recent generations. 

The participant numbers described in the first two sentences of the results section need to be more clearly explained, as not clear how the final participation numbers are derived. 

There is a typo on line 151

Can you be more specific than ‘Heart’ in Table 3

You state this on the tables: ‘Note: Difference statistically significant as denoted by non-over-lapping 95% confidence intervals (Bold).’ Are any supposed to be in bold?

Some of the tables need formatting as some subheadings are not aligned

In Table S5, is age 70-74 in Model 3 meant to be bold, as stated in main text that there is no significance?

The sentence on line 201 needs to be reworded to be more clear

On line 205, there is a note to add a citation

Reference formatting is needed on line 230

Author Response

Reviewer 1

  1. This is an interesting read, although concerning that incidence and progression of chronic periodontitis is increasing with more recent generations. 

Thanks

  1. The participant numbers described in the first two sentences of the results section need to be more clearly explained, as not clear how the final participation numbers are derived. 

We have re-written the first two sentences (Last paragraphs, page 6)  

  1. There is a typo on line 151

We have deleted ‘a’ in the sentence. (Last paragraph, page 6)

‘the two cohorts, with more than half in the younger age group (60-69 years) and…’

  1. Can you be more specific than ‘Heart’ in Table 3

We have changed to ‘Heart disease’ (Tables 3-4)

  1. You state this on the tables: ‘Note: Difference statistically significant as denoted by non-over-lapping 95% confidence intervals (Bold).’ Are any supposed to be in bold?

Thanks. Yes, we have highlighted that the statistically significant differences pertain to the estimates in the tables 2 and 3.

  1. Some of the tables need formatting as some subheadings are not aligned

We have re-formatted the subheading (Table 2, and Table 4)

  1. In Table S5, is age 70-74 in Model 3 meant to be bold, as stated in main text that there is no significance?

Yes, you are right, we have re-written the sentences in the text (Third paragraph, page 9)

  1. The sentence on line 201 needs to be reworded to be more clear

We have re-written the sentence (Fourth paragraph, page 8)

  1. On line 205, there is a note to add a citation

Thanks, we have deleted the note. (Fourth paragraph, page 8)

  1. Reference formatting is needed on line 230

We have re-formatted reference #18 (Last paragraph, page 10)

Reviewer 2 Report

The paper is well designed and with many important variables being analyzed. A single point that would deserve a better description would be the discussion of the study. As several factors were raised as influencers in such a pathology analyzed, why not discuss it further? In this reasoning, still in view of the importance of the conclusions observed, what to improve in future periodontal treatments so that the rate of the disease decreases in other patients?

Author Response

Reviewer 2

The paper is well designed and with many important variables being analyzed. A single point that would deserve a better description would be the discussion of the study. As several factors were raised as influencers in such a pathology analyzed, why not discuss it further? In this reasoning, still in view of the importance of the conclusions observed, what to improve in future periodontal treatments so that the rate of the disease decreases in other patients?

Thanks, we have added more related content and evidence in the Discussion section (Page 9-10)

Reviewer 3 Report

Overall the manuscript is well written. However, a few corrections need to be addressed.

The discussion section of the article requires revision where more literature needs to be discussed by correlating the current study results with the previously performed studies.

Discussion part of the article seems to be small and needs to be expanded in a more elaborate manner.

Page 9, line number 205: Appropriate references needs to be given in the manuscript.

Author Response

Reviewer 3

  1. Overall the manuscript is well written. However, a few corrections need to be addressed. The discussion section of the article requires revision where more literature needs to be discussed by correlating the current study results with the previously performed studies. Discussion part of the article seems to be small and needs to be expanded in a more elaborate manner.

Thanks, we have added more related content and evidence in the Discussion section to fill out our discussion (Page 9-10)

  1. Page 9, line number 205: Appropriate references needs to be given in the manuscript.

Thanks, we have deleted the note ‘add a citation’ (Fourth paragraph, page 9)

Reviewer 4 Report

This paper is very important to reflect periodontal disease prevalence in Australia. There is more data on this from US and Europe but not much from Australia. However, this paper should be reworked to include the 2017 World Workshop classification terminology. Most of the terms used in the paper are from 1999 AAP classification and is not in use anymore in US/Canada. For a further international reach, I would recommend this should be reworked. No further changes are recommended

Author Response

Reviewer4

This paper is very important to reflect periodontal disease prevalence in Australia. There is more data on this from US and Europe but not much from Australia. However, this paper should be reworked to include the 2017 World Workshop classification terminology. Most of the terms used in the paper are from 1999 AAP classification and is not in use anymore in US/Canada. For a further international reach, I would recommend this should be reworked. No further changes are recommended

Thanks, we have added related context and terms in future research (First paragraph, page11)